# Evaluation of Spatiotemporal Patterns of the Spinal Muscle Coordination Output during Walking in the Exoskeleton

**DOI:** 10.3390/s22155708

**Published:** 2022-07-30

**Authors:** Dmitry S. Zhvansky, Francesca Sylos-Labini, Arthur Dewolf, Germana Cappellini, Andrea d’Avella, Francesco Lacquaniti, Yury Ivanenko

**Affiliations:** 1Institute for Information Transmission Problems, Russian Academy of Sciences, 127994 Moscow, Russia; dmitry.zhvanskij@phystech.edu; 2Laboratory of Neuromotor Physiology, IRCCS Santa Lucia Foundation, 00179 Rome, Italy; f.syloslabini@hsantalucia.it (F.S.-L.); arthur.dewolf@uclouvain.be (A.D.); g.cappellini@hsantalucia.it (G.C.); a.davella@hsantalucia.it (A.d.); lacquaniti@med.uniroma2.it (F.L.); 3Department of Systems Medicine, University of Rome Tor Vergata, 00133 Rome, Italy; 4Department of Biomedical and Dental Sciences and Morphofunctional Imaging, University of Messina, 98100 Messina, Italy; 5Department of Systems Medicine and Center of Space Biomedicine, University of Rome Tor Vergata, 00133 Rome, Italy

**Keywords:** walking, muscle coordination, spinal locomotor output, body unloading, exoskeletons, benchmarking

## Abstract

Recent advances in the performance and evaluation of walking in exoskeletons use various assessments based on kinematic/kinetic measurements. While such variables provide general characteristics of gait performance, only limited conclusions can be made about the neural control strategies. Moreover, some kinematic or kinetic parameters are a consequence of the control implemented on the exoskeleton. Therefore, standard indicators based on kinematic variables have limitations and need to be complemented by performance measures of muscle coordination and control strategy. Knowledge about what happens at the spinal cord output level might also be critical for rehabilitation since an abnormal spatiotemporal integration of activity in specific spinal segments may result in a risk for abnormalities in gait recovery. Here we present the PEPATO software, which is a benchmarking solution to assess changes in the spinal locomotor output during walking in the exoskeleton with respect to reference data on normal walking. In particular, functional and structural changes at the spinal cord level can be mapped into muscle synergies and spinal maps of motoneuron activity. A user-friendly software interface guides the user through several data processing steps leading to a set of performance indicators as output. We present an example of the usage of this software for evaluating walking in an unloading exoskeleton that allows a person to step in simulated reduced (the Moon’s) gravity. By analyzing the EMG activity from lower limb muscles, the algorithms detected several performance indicators demonstrating differential adaptation (shifts in the center of activity, prolonged activation) of specific muscle activation modules and spinal motor pools and increased coactivation of lumbar and sacral segments. The software is integrated at EUROBENCH facilities to benchmark the performance of walking in the exoskeleton from the point of view of changes in the spinal locomotor output.

## 1. Introduction

In the last decades, lower limb exoskeleton robotic devices have been developed for various purposes, including gait assistance and the rehabilitation of patients [1]. For instance, to provide patients with some degree of locomotion capability, passive unpowered orthoses are often prescribed [2], though these devices have limitations, including the high energy expenditure and low utilization by individuals with severe gait impairments. Powered exoskeletons are extensively developed to provide new possibilities for severely paralyzed patients to walk [1,3,4,5,6,7,8]. In addition to medical applications for gait assistance and rehabilitation in patients, exoskeletons are also being developed for other applications for healthy subjects (military, industry, agriculture, etc.). Many of these devices include some form of body weight support and adjustable levels of robotic guidance forces.

The development of exoskeletons and new control implementations, including brain-machine interfaces, can benefit from appropriate procedures and metrics for their evaluation [9,10]. Various indicators of gait performance in the exoskeleton can be used based on kinematic/kinetic measurements, such as maximal speed, travelled distance, range of angular motion, joint torques, symmetry of lower limb movements, cognitive effort, etc. Generic kinematic and kinetic indicators seem to prevail, in spite of the metrics of human–robot interaction [8]. While such variables provide general characteristics of gait performance, only limited conclusions can be made about the neural control strategies based on these characteristics. Furthermore, some kinematic or kinetic parameters are a consequence of the control implemented on the exoskeleton. Therefore, standard indicators based on kinematic variables have limitations and need to be complemented by performance measures of muscle coordination and control strategy such as those proposed by our approach.

In recent years, many researchers put significant efforts into assessing the functional output of the spinal locomotor circuits in humans [11]. The principles that the central nervous system uses to govern hundreds of muscles to control whole-body movements include a modular organization of the neuronal networks [12]. In particular, functional and structural changes at the spinal cord level induced by exoskeleton assisted walking can be mapped into muscle synergies and spinal maps of motoneuron (MN) activity, as a means to look backward from the periphery to the central motor programming. Decoding the spinal locomotor output and assessment of the spatiotemporal muscle activity patterns, as indicators of motor function, have become an essential tool for investigating the function of pattern generation networks in the spinal cord [13,14,15,16,17,18,19,20]. Additionally, considerable changes in the muscle coordination output might occur with body unloading and during walking with the exoskeleton [21,22,23]. An abnormal spatiotemporal integration of activity in specific spinal segments may result in a risk for failure or abnormalities in gait recovery [17,24]. There is also a differential involvement of spinal motoneuronal and interneuronal circuits in different locomotor tasks [24,25]. Therefore, this new information and the corresponding benchmarking performance indicators are much needed in the context of locomotor adaptation and impairments, evaluating the effect of exercise while walking in the exoskeleton and spinal plasticity.

Here we present the PEPATO software (SW), which is a benchmarking solution to assess changes in the spinal locomotor output during walking in the exoskeleton. The proposed outcomes may provide important information about changes in the neural control strategy and spinal locomotor output, that will complement other performance indicators and enrich evaluation capabilities of wearable exoskeletons and their users. First, we describe the protocol, the SW pipeline, and the initial data pre-processing (Section 2.1.1 and Section 2.1.2). The detailed description of the performance indicators is provided in Section 2.1.3 (motor modules) and Section 2.1.4 (spinal maps). Finally, we present an example of usage of this SW (Section 2.2 and Section 3) for evaluating walking in an unloading exoskeleton [26] that allows a person to step in simulated reduced (the Moon’s) gravity. The reason for using the unloading exoskeleton was to demonstrate the wide variety of tasks in which the SW can be applied. Simulated reduced gravity represents a well-controlled technique to study body weight unloading and is widely used in gait rehabilitation as a tool to facilitate locomotor activity in individuals with neuromotor disorders, such as spinal cord injury, stroke, Parkinson disease, multiple sclerosis, cerebral palsy, etc. [27].

## 2. Materials and Methods

### 2.1. PEPATO Software for Evaluating Changes in the Spinal Locomotor Output during Walking in the Exoskeleton

The proposed software provides performance indicators (PI) of muscle coordination and spinal locomotor output based on EMG signals. It generates the two main groups of PIs related to muscle modules and spinal maps of motoneuron activity. Only offline processing is available; the SW is not configured for real-time analysis. The general scheme of the EMG data processing for walking with the exoskeleton is illustrated in Figure 1.

The rationale for the proposed performance indicators is the following. First, muscle synergies during walking in the exoskeleton may demonstrate significant changes in both synergy weights and synergy temporal activations with increasing exoskeleton work and torque in unimpaired individuals [28,29]; the center of activity also shows significant changes [21] along with a non-linear scaling and reorganization of EMG activity [21,22,23]. Second, specific changes in the modular organization of muscle patterns occur after neurological lesions [30,31,32]. Patients often exhibit decreased neuromuscular complexity during gait [30,33], demonstrate impaired muscle synergies and temporal activation patterns [30,31,32,34] and may increase neuromuscular motor module consistency following rehabilitation [35]. Spinal functional topography can be assessed by mapping multi-muscle EMG patterns onto the rostrocaudal location of the spinal MN pools and provides important information about pattern generator output during locomotion in terms of segmental control rather than in terms of individual muscle control [36]. Widening of spinal segmental output and alterations in the relative activation timing of sacral and lumbar motor pools represent important physiological markers of pathological gaits [17,24,37,38] and age-related changes [15,34,39]. Below we describe in detail the protocol and proposed benchmarking performance indicators to assess changes in the spinal locomotor output during walking in the exoskeleton.

#### 2.1.1. Protocol, Input, Reference Data, and User Interface

Protocol

The protocol consists in walking on a treadmill at a constant speed: 2, 4, and 6 km/h. This set of speeds evenly covers the common range of normal human walking speeds, from a slow walk (2 km/h) to almost a run (6 km/h), and thus can serve as a reference for exoskeleton walking. At least 10 consecutive strides should be recorded for each speed condition. PEPATO software allows to characterize only one or two speed conditions in case the participant cannot perform all speed conditions (e.g., some exoskeletons or their usage by patients allow only slow walking).
Input-Electromyographic (EMG) activity recorded unilaterally from lower limb muscles during walking in the exoskeleton. The current version of PEPATO processes EMG activity of 8 muscles (unilaterally or bilaterally) that represent flexor and extensor groups of both distal (shank) and proximal (thigh) lower limb segments and are most accessible for recordings during walking in most exoskeletons: soleus (Sol), gastrocnemius medialis (GaLa), tibialis anterior (TiAn), rectus femoris (ReFe), vastus lateralis (VaLa), vastus medialis (VaMe), semitendinosus (SeTe), biceps femoris (long head, BiFe). This set of muscles represent major extensor and flexors groups of muscles controlling walking and are often used in both simulation studies and when evaluating basic muscle modules [40,41,42]. Also, these muscles constitute a relatively large part of the total cross-sectional area of lower limb muscles [43]. The PEPATO SW can handle a lower (see the below experiment with 7 bilateral muscles that include flexors and extensors of the thigh and shank segments) or larger number of muscles (see below REFERENCE DATA).-Gait events for the stride identification that specify the timing of touchdown events of the recorded strides.Reference Data

For initial benchmarking of the PEPATO SW, we created two sets of reference data according to the specifications provided below. However, future users of the SW may want to create their own set of reference data based on the specific characteristics of the population and conditions to be tested, e.g., age range, sex, typicality or pathology, normal walking or walking with an exoskeleton, speed range, etc.

Reference data consist of EMG data and corresponding reference performance indicators for comparing muscle coordination output with that of neurologically intact individuals during normal walking without an exoskeleton on a treadmill at 2, 4, and 6 km/h. Each walking trial lasted ~40 s and was performed on a standard treadmill. The intervals between trials were ~1 min. Prior to data collection, the subjects were trained on each task, allowing them time to familiarize with the various walking conditions. EMG data were recorded at 2000 Hz using a Delsys Trigno wireless system (Boston, MA, USA). Kinematic data were recorded at 100 Hz by means of a Vicon-612 system (Oxford, UK) with nine cameras. Sampling of kinematic and EMG data were synchronized. For each trial, at least 10 full gait cycles were recorded, and no more than 10% of them were dropped due to the artefacts of different nature (evoked by movement, electrode contact interruption). Since the differences related to the operator’s age had been observed [15,44], a reference database was obtained for the two groups of adults: young (*n* = 10 subjects, age range 19.9–36.4 years) for all speeds (2, 4, and 6 km/h) and elderly (*n* = 10 subjects, age range 65.0–80.1 years) for 2 and 4 km/h. Aging adults (65–80 years) may also serve as a control group for the populations of patients with an older age (e.g., Parkinson disease, stroke, etc.). Participants ranging from 40 to 65 years were not included, in order to sharpen the age differences between the two groups, however, the PEPATO SW allows data reference updating. These data and corresponding performance indicators are available at https://github.com/dzhvansky/pepato (accessed on 28 May 2022). In the PEPATO software, the reference database serves to generate directly P1.5 and P2.4 performance indicators (Figure 1) for the individual subjects walking in the exoskeleton, as well containing other PIs for the subjects in the reference database that can be used for the subsequent statistical analysis performed by the user to compare the group of participants walking in the exoskeleton with the reference group.

User Interface

Software was developed using Matlab (MathWorks Inc., Natick, MA, USA). It is integrated at EUROBENCH facilities (https://eurobench2020.eu/, accessed on 28 May 2022), which is a benchmarking solution to produce assessments of changes in the spinal locomotor output during walking in the exoskeleton with respect to reference data on normal walking. Input and output data for the PEPATO software are compliant with the EUROBENCH data format. PEPATO allows data reference updating to include more muscles, subjects, and conditions. The software can be used also for other scenarios (directly for all indicators that do not depend on the reference data or including the reference data on the additional scenarios for indicators that depend on the reference data).

#### 2.1.2. Data Pre-Processing

Data pre-processing is an integral part of the PEPATO software and serves not only to upload the data and provide a necessary segmentation of EMG envelopes across several gait cycles, but also to perform a quality check of the EMG data. The main steps of the data pre-processing chain are illustrated in Figure 2a and contain: parameter selection, data loading, initial data analysis and visualization of EMGs along with gait cycle timing and EMG spectra for visual inspection, exclusion of gait cycles with potential artefacts, exclusion of muscles with artefacts, identification of spectrum-related artefacts and their correction, warning for potential artefacts based on advanced artefact detection in individual cycles and their exclusion, EMG rectification, and low-pass filtering.

The raw EMG signals are initially band-pass (30–400 Hz) filtered using a zero-lag filter and visualized on the screen along with the timing of gait touchdown events and Fourier spectra. The ‘parameter selection’ option includes a confirmation (or a change) of some default parameters and also a selection of the method that can be used for the selection of the number of motor modules (see below). A segmentation of the gait cycles during this visualization phase is particularly useful for a subsequent identification of cycles with potentially noisy EMGs.

The quality check of the EMG channels is performed in order to warn for a potential presence of noise and to improve the EMG data processing. Instrumental constraints and artefacts of EMG data recording may be related to the difficulties in placing EMG electrodes ‘safely’ on some muscles while walking in some types of the exoskeletons or due to the vicinity of EMG sensors to the powered motors of the exoskeleton. Software allows (after a confirmation by the user) to exclude gait cycles with suspicious muscle activity and correct some types of artefacts. In particular, frequency-related artefacts may take place (e.g., 50-Hz) when using some EMG sensors (e.g., see “spectrum-related artefact detection” in Figure 2b with an example of the presence of simultaneous peaks in all EMG channels at ~155-Hz frequency) and can be automatically filtered using the notch filter. Visual inspection also allows to detect and exclude gait cycles with ‘evident’ artefacts (e.g., due to a temporal lack/saturation of the signal following EMG sensor detachment or its direct contact with the metal part of the exoskeleton). Finally, we also implemented an advanced detection of potential ‘artefacts’ in individual gait cycles to suggest candidates for visual inspection. Individual cycle can be considered as an ‘outlier’ if its correlation coefficient with the ensemble-averaged EMG envelope across all gait cycles is less than 0.6. It is recommended nevertheless, prior to the experiments, to ensure the best placement of the EMG electrodes and quality of data recording in order to minimize potential artefacts since the PEPATO software provides an automatic detection and correction of some types of artefacts but it cannot ‘restore’ the original EMG signals.

Following data pre-processing, the EMG signal from each muscle is full-wave rectified, low-pass filtered using a fourth-order (zero-phase) Butterworth filter with a cut-off of 10 Hz, and time interpolated to 200 points for each individual gait cycle to be used for the assessment of the spinal locomotor output (motor modules and spinal maps, see below).

#### 2.1.3. Performance Indicators for the Assessment of Motor Modules

For the assessment of motor modules, the software algorithms evaluate both spatial (muscle weightings or synergies) and temporal (basic patterns) components of muscle modules. The amplitude of each rectified and low-passed EMG signal is normalized to its maximum value across all strides and all speed conditions for each subject.

Motor modules are extracted using the NNMF algorithm, as previously described [34,45]. Briefly, the processed EMG envelopes were combined into an *m* × *t* EMG matrix (where *m* is the number of muscles, and *t* is the number of gait cycles × 200). Then, the NNMF algorithm extracts the basic activation patterns (*P*) and weighting coefficients (*W*) by searching for an approximate solution of the root-mean-squared *error* between the EMG matrix and *W* × *P*, according to the formula:(1)$$\mathrm{EMG}=\sum_{i}^{n}P_{i}W_{i}+error,\underset{}{}n\leq m$$

Several criteria widely used in the previous studies are incorporated into the software algorithms and can be chosen (‘parameter selection’), namely: the ‘best linear fit’ method based on the percent of variance accounted for, VAF [45,46], using the threshold (for instance, 90%) to account for the total variance in EMG data [20,42], or fixing the number of locomotor modules to 4. The rationale for the latter criterion is that the number of basic locomotor modules reported in numerous studies with a reduced number of recorded muscles is typically four [13,41]. For the best linear fit, a method is based on a linear regression procedure [46] by varying the number of basic patterns from 1 to 8 and selecting the smallest *n* such that a linear fit of the VAF vs. *n* curve had a residual mean square error <10^−4^.

The following performance indicators were chosen to characterize motor modules: P1.1, the number of motor modules, P1.2, reconstruction quality (R^2^) of EMG patterns from the number of motor modules in P1.1, P1.3, FWHM (full width at half maximum)—duration estimate of basic patterns, P1.4, centre of activity (*CoA*) of basic patterns, P1.5, the degree of similarity with the reference group.

Pattern decomposition (P1.2) is characterized by calculating the total variance accounted for (VAF or R^2^) (Figure 2c, left panel) [47]:VAF = sum of squared errors/total sum of squares (2)
where the total sum of squares is taken with respect to the mean over the rows of the data matrix.

The FWHM (P1.3) is calculated as the sum of the durations of the intervals in which the basic activation patterns exceeded half of its maximum (Figure 2c, right panel) [45].

The CoA (P1.4) characterizes the phase of basic activation patterns with respect to the gait cycle and was calculated using circular statistics [48]. The CoA of the EMG waveform is calculated as the angle of the vector (first trigonometric moment) which points to the center of mass of that circular distribution using the following formulas:(3)$$A=\overset{200}{\underset{t=1}{\sum }}\left(\cos\theta_{t}\times {\mathrm{EMG}}_{t}\right)\;$$
(4)$$B=\overset{200}{\underset{t=1}{\sum }}\left(\sin\theta_{t}\times {\mathrm{EMG}}_{t}\right)$$
(5)$$\mathrm{CoA}=\tan^{-1}\left(B/A\right)$$

The CoA was chosen because it was impractical to reliably identify a single peak of activity in some of the basic patterns, especially in patients [21,32]. It can only be considered as a qualitative parameter, because averaging between distinct foci of activity may lead to misleading activity in the intermediate zone. Nevertheless, it can be helpful to understand if the distribution of muscular activity remains unaltered across different groups.

The FWHM and CoA were calculated for all motor modules in each participant. To order, analyse, and average the modules across participants, we used the degree of their similarity with the reference group (P.1.5), assessed the characteristics of similar motor modules, and identified unmatched motor modules (Figure 2a). A locomotor module is considered as a functional unit implemented in a neuronal network of the spinal cord to generate a specific spatiotemporal structure to muscle activations [49,50]. It involves both a basic activation pattern (temporal structure) with variable weights of distribution (spatial structure) to different muscles. Therefore, to identify and average similar motor modules across participants and compare with the group of control (reference) subjects, the degree of similarity is evaluated based on the best matching of both W and P using the cluster analysis similar to that reported in our previous studies [34,45]. Cluster membership for modules is defined as follows. In a 2 × *m* dimensional space (*m* features for muscle weights and *m* features for basic temporal patterns [to this end, basic patterns are time interpolated to *m* points]) all features are Z-normalized; after subtracting the mean, each feature is divided by the standard deviation. Clusters are calculated in this space using the k-means algorithm. A module is considered to belong to a cluster if the average (for all features) Euclidean distance to the centre of the cluster does not exceed 0.8, and the maximum Euclidean distance does not exceed 2. If motor modules were not matched to the modules from the reference set, we isolated those unmatched (non-classified) modules. In addition to associating motor modules with reference clusters (reference modules) for the subsequent analysis and averaging across participants, the algorithms also provide similarities of both basic patterns and muscle weights with the references set; the Pearson correlation coefficient was used for patterns, and the normalized scalar product for weights.

#### 2.1.4. Performance Indicators for the Assessment of the Spinal Maps of Motoneural Activation

To assess the spatiotemporal organization of the spinal locomotor output, the EMG-activity profiles are mapped onto the rostrocaudal anatomical location of MN-pools in the human spinal cord, as previously described. This method can be used to characterize the lumbosacral motor pool activation by considering relative intensities, and spatial and temporal structure of the spinal segmental locomotor output [14,15,17,36,37,51,52,53]. Briefly, each lumbosacral segment supplies several muscles, and each muscle is innervated by several spinal segments. Even if the number of recorded muscles during walking in the exoskeleton is limited, we have previously demonstrated that these muscles contribute largely to the overall spinal maps [39,51]. Moreover, the recorded set of muscles constitute a relatively large part of the total cross-sectional area of lower limb muscles [43]. To reconstruct the MN output pattern of any given (L2–S2) spinal segment, we subdivided each segment into six slices, according to the myotomal charts of Sharrard [54], resulting in 36 subsegments *S_j_*. Then we applied the following formula [51]:(6)$$S_{j}=\frac{\sum_{i=1}^{m_{j}}{\mathrm{EMG}}_{i}}{m_{j}}\;$$
where EMG*_i_*—the participant-specific envelope of the *i*-th muscle activity (in mV), and *m_j_*—the number of muscles innervated by the *j*-th subsegment. To visualize a continuous smoothed rostro-caudal spatiotemporal activation of the spinal cord [36], we used a filled contour plot that computes isolines calculated from the activation waveform matrix (36 slices × 200 points) and fills the areas between the isolines using constant colors (the contourf.m function in Matlab). Using the Sharrard myotomal charts [54], the resulting spatiotemporal spinal map consisted in thirty-six discrete rostrocaudal activation waveforms (since the anatomical data of the Sharrard chart are broken down into thirty-six slices), but by averaging the activity of six slices in each segment we could reconstruct the MN activation of each spinal segment to provide the performance indicators for the spinal maps of MN activity.

To compare the relative activation of the lumbar and sacral segments in each walking condition, we averaged the motor output patterns over the gait cycle in the upper part of the lumbar (sum of the activity from L3 to L4) and the sacral segments (sum of activity from S1 to S2). To reduce overlaps due to maps smoothing, the spinal segment L5 was not considered, and the segment L2 was also not considered, as it contains considerably smaller number of MNs than L3 and L4 [55].

The following performance indicators are chosen to characterize the spinal maps of MN activity: P2.1, timing of maximum activation of sacral (S1 + S2) and upper lumbar (L3 + L4) motor pools; P2.2, FWHM of activation of sacral (S1 + S2) and upper lumbar (L3 + L4) spinal motor pools; P2.3, co-activation index (CI) of sacral and upper lumbar motor pools; and P2.4, the degree of similarity (correlation) of activation of sacral and lumbar motor pools with respect to the reference group.

To report the general spatiotemporal characteristics of the spinal maps, we calculated the timing of the maximal activation (P2.1) and the FWHM (P2.2) throughout the gait cycle. The FWHM was computed in the same way as for the motor modules (see above).

The co-activation index (P2.3) was assessed between the lumbar (L3 + L4) and sacral (S1 + S2) segments using the following formula [34,56]:(7)$$CI=\frac{\sum_{j=1}^{200}\left[\left({\mathrm{EMG}}_{H}\left(j\right)+{\mathrm{EMG}}_{L}\left(j\right)/2\right)\right]\times \left({\mathrm{EMG}}_{L}\left(j\right)/{\mathrm{EMG}}_{H}\left(j\right)\right)}{200}$$
where EMG*_H_* and EMG*_L_* represent the highest and the lowest activity between the pairs (for calculating the *CI*, activity of the lumbar and spinal segments is normalized to its maximum value). In order to have a global measure of the co-activity level, the *CI* was then averaged over the entire gait cycle (*j* = 1:200). This method provides a sample-by-sample estimate of the relative activation of the upper lumbar and sacral segments as well as the magnitude of the co-contraction over the entire cycle. Using this equation, high co-contraction values represent a high level of activation, whereas low co-contraction values indicate either low level activation of these segments, or a high level activation of one along with low level activation of another one in the pair [56].

### 2.2. Usage of the PEPATO Software for the Assessment of Walking in the Unloading Exoskeleton

#### 2.2.1. Participants

Nine healthy volunteers (age range between 25 and 45 years, 7 males and 2 females, height 179.4 ± 6.2 cm, weight 75.7 ± 11.1 kg [mean ± SD]) participated in the study. The study conformed to the Declaration of Helsinki, and informed consent was obtained from all participants according to procedures approved by the Ethics Committee of the Santa Lucia Foundation (Prot. CE-PROG.274-22).

#### 2.2.2. Unloading Exoskeleton

The unloading exoskeleton as an integral part of the tilting body weight support (BWS) system (manufactured by RTC, Rome, Italy) was constructed to simulate more realistic effects of gravity changes on both the stance and swing of legs in the sagittal plane (Italian patent no. Rm2007A000489). The detailed description of the unloading exoskeleton and tilted body weight support system is provided elsewhere [26,57,58]. Briefly, participants lay on their right side with each leg suspended in the exoskeleton (Figure 3b, left panel), allowing low-friction joint rotation due to bearing junctions. The length of the telescopic thigh segment of the exoskeleton was adjusted according to the subject’s thigh length, and the leg was attached to the exoskeleton (fastened by a cuff) to provide the best alignment of the axes of rotation of the knee and hip joints with those of the exoskeleton. The foot segment remained unrestrained in air. The upper body of the subject was secured through a chest and shoulder fixation, with the head placed on the pillow roller. Tilted BWS more realistically simulates the downward force acting on both the center of mass (COM) and swinging limbs [26,57,58] but prevents arm oscillations, adds inertia (15 kg chassis, 3 kg exoskeleton), and somewhat limits trunk movements in the anterior–posterior direction. The construction of the tilting unloading exoskeleton is based on the idea of neutralizing the component of the gravity force normal to the lying surface [mg·cos (α), where α is the angle of inclination] [59], while the component of the gravity force acting on the body and swinging limbs in the sagittal plane is reduced in relation to the tilt angle [mg·sin (α)]. The participant stepped on the treadmill that was tilted to the same angle (Figure 3b, left panel). The apparatus permitted low-friction up-and-down (relative to the treadmill) sliding of the supporting chassis over two parallel tracks formed by a steel beam. Although anterior–posterior trunk movements were limited, the hip support could slide along the anterior–posterior guides of the couch, thus allowing pelvis rotations. In this experiment, the participants walked at 4 km/h and the BWS level was set to simulate the Moon’s gravity (16% of body weight).

#### 2.2.3. Data Recording and Processing

We recorded kinematic data bilaterally at 100 Hz by means of the Vicon-612 system (Oxford, England) with nine video cameras spaced around the walkway. Infrared reflective markers (diameter 1.4 cm) were attached on each side of the subject to the skin overlying the following landmarks: greater trochanter (GT), lateral femur epicondyle (LE), lateral malleolus (LM), heel (HE), and fifth metatarso–phalangeal joint (5 MP). The spatial accuracy of the system was better than 1 mm (root mean square). The GT marker of the right side of the body could not be recorded (as the subject laid on the right side); additionally, the GT and LE landmarks of the left leg were recorded by attaching the 20 cm sticks with two markers to the appropriate joint and the GT and LE positions were reconstructed as a midpoint between these two markers.

EMG activity of the Sol, GaLa, TiAn, ReFe, VaMe, SeTe, and BiFe muscles was recorded bilaterally at 2000 Hz by means of surface electrodes using the wireless Zerowire system (Aurion, Milan, Italy). The recording system bandwidth was 20–1000 Hz with an overall gain of 1000. Sampling of kinematic and EMG data were synchronized.

Gait cycle was defined as the time between two successive foot contacts by the same leg according to the local minima of the ‘vertical’ displacement of the HE marker [58] and the timing of these gait events and EMG data were saved in the format adopted by the PEPATO SW for the subsequent analyses.

#### 2.2.4. Statistics

The experimental data set mostly did not meet the normal distribution criteria (the Shapiro–Wilk W-test, *p* < 0.05), therefore non-parametric statistics were used for statistical data analysis. Between-groups differences in the motor modules and spinal maps parameters were assessed by performing the Mann–Whitney tests. For paired comparisons of two related groups, the Wilcoxon matched-pairs test was used. The analysis of CoA and the timing of maximum activation for the spinal maps was performed using the Watson–Williams test for circular data [60]. Multiple testing correction was assessed according to the Benjamini–Hochberg procedure. Since there were no significant differences between the data of the left and right body sides, the performance indicators were averaged between both sides of the body. Descriptive statistics included medians, quartiles, and range of values. Significance level was set at *p* < 0.05.

## 3. Results

### 3.1. EMG Activity during Walking in the Unloading Exoskeleton

Figure 3 illustrates an example of EMG activity of lower limb muscles in two participants walking at 4 km/h: one subject from the reference group (panel a) and one subject who walked in the unloading exoskeleton (panel b). The timing of the touchdown events was also depicted (vertical lines) to separate the individual gait cycles. Generally, the cycle duration was longer during walking in the unloading exoskeleton when compared at the same speed, in accordance with previous studies on the effect of body unloading [58].

Consistent with reduced muscle efforts during walking in the simulated Moon’s gravity, overall there were considerable changes (decrease) of the amplitude of EMG activity of all tested lower limb muscles, both extensors and flexors, with respect to normal upright walking (Figure 3). Since there could also be very low activity in some muscles in some individuals, especially at low speeds (e.g., <3 km/h, see [22]), we performed these comparisons at a relatively high speed 4 km/h. Higher speeds were not considered since locomoting at speeds >4 km/h in the unloading exoskeleton corresponds to running gait rather than walking [26,57,58]. During walking in the exoskeleton at 4 km/h, all subjects demonstrated prominent, even though smaller, EMG activity. Figure 3a,b illustrates both similarities in the EMG patterns and some differences between these two subjects (e.g., in the activity of flexor muscles, such as BiFe, SeTe, TiAn) and some changes in the timing of EMG activity bursts. Changes in the spatiotemporal structure of the spinal locomotor output during walking in the unloading exoskeleton were then evaluated using the PEPATO software.

### 3.2. Performance of the Benchmarking Software for Evaluating Motor Modules during Walking in the Unloading Exoskeleton

The PEPATO software successfully generated the performance indicators for the assessment of the motor modules and spinal maps of MN activity for each participant. If during the execution of a quality check, the EMG data did not comply with the pre-defined tolerance ranges for the absence of artefacts, the system detected them and provided the corresponding visual feedback to the user suggesting to exclude specific EMG channels or strides. In particular, after the quality check of the EMG data, overall ~5% of gait cycles were excluded mainly due to the presence of movement-related burst ‘outliers’, and one participant’s data were excluded due to the high percentage (>50%) of cycles and EMGs treated as irrelevant. The data of remaining eight participants were successfully processed for extracting motor modules and spinal maps features.

The results of the analysed EMG activities showed that all four motor modules of the reference data were also present in most participants walking in the unloading exoskeleton. The total VAF for EMG data reconstruction was similar for the two groups of subjects; more than 90% of variance was accounted for by four basic patterns in each group (R^2^ > 0.9, Figure 4d, left panel). However, the analyses also revealed significant differences with the reference data.

First, using NNMF and a cluster analysis (see Materials and Methods), we determined the percentage of motor modules in participants walking in the unloading exoskeleton not associated with any of the reference data modules (statistics for both sides of the body were pooled together): 14% for the 1st module, 14% for the 2nd module, 7% for the 3rd module, and 33% for the 4th module. Thus, the 4th motor module was the least common for participants walking in the exoskeleton.

Second, several characteristics (performance indicators) of the common motor modules significantly differed between the two groups. Ensemble-averaged basic temporal patterns and corresponding muscle weights are plotted for each group in Figure 4a. Muscle weights were rather similar although some relatively small differences were present for the 2nd module (VaMe *p* = 0.0007, ReFe *p* = 0.02) and 4th module (ReFe *p* = 0.007, GaLa *p* = 0.01) (Figure 4a, right column). Similarities in the muscle weights assessed by using the scalar products were comparable across modules despite some trend visible in Figure 4c (lower panel) and reflecting some variability for the 4th module. Basic temporal patterns showed more differences from those of the reference data. Similarities in the basic patterns assessed by using correlations also showed some trends and a more variable pattern for the 4th module (Figure 4c). The CoA demonstrated some differential changes for two modules (while other modules did not differ): a shift of the CoA toward the midstance (later timing) for the 1st module (*p* = 0.02) and toward early timing for the 3rd module (*p* = 0.0016) with respect to the reference data (Figure 4d, middle panel). The FWHM differed significantly and again showed differential changes depending on the motor module, being smaller for the second basic temporal pattern (*p* = 0.015) and larger for the third and fourth basic patterns (*p* = 0.003 and *p* = 0.009, respectively) (Figure 4d, right panel).

### 3.3. Performance of the Benchmarking Software for Evaluating Spinal Maps

The PEPATO algorithms revealed specific changes in the spinal maps of alpha MN activity approximated by mapping the EMG-activity profiles onto the rostrocaudal anatomical location of MN-pools in the human spinal cord. Figure 5a illustrates the examples of the spatiotemporal patterns of MN activity in the lumbosacral enlargement obtained from the subject walking normally (left panels) and the subject walking in the exoskeleton (right panels). Various performance indicators were evaluated, including potential differences between sacral and upper lumbar segment activations (see Materials and Methods), and plotted in panel b. Similarity between the two groups of participants was higher for the lumbar segment activation than for the sacral segment activation (*p* = 0.016, Figure 5b, left panel). Although no significant differences were observed in the peak activity timing, it tended to be more variable for the group of subjects walking in the exoskeleton (Figure 5b, second left panel). Other performance indicators differed significantly (Figure 5b, right panels). In particular, the FWHM was smaller for the sacral segments (*p* = 0.019), but larger for the upper lumbar segments (*p* = 0.019), and the treated group demonstrated significantly higher co-activation of sacral and lumbar segments (*p* = 0.03).

## 4. Discussion

An objective assessment of adaptation to walking in an exoskeleton was realized by developing the benchmarking PEPATO software for processing the EMG activity recordings from the lower limb muscles obtained throughout a predefined protocol of walking at specific speeds (Figure 1 and Figure 2). The data acquired were saved, classified, and used for a subsequent analysis based on synthetic metrics and performance indicators to detect differences in the spinal motor output during walking in the exoskeleton with respect to normal walking. The proof of principle for using this software is provided by analyzing the spinal locomotor output and muscle coordination patterns during walking in an unloading exoskeleton that allows a person to step in simulated reduced (the Moon’s) gravity by tilting the body relative to the vertical (Figure 3, Figure 4 and Figure 5).

Current approaches and available methodology to evaluate muscle coordination patterns during locomotion are mainly based on selected and limited parameters of multi-muscle activity and, to our knowledge, there is no SW that comprehensively assesses both muscle modules and spinal maps of MN activation. For example, the SW muscle synergies [61] offers a complete analysis framework of muscle synergies addressed to scientists and more advanced users, with customizable sensible defaults depending on the specifics of the study design, but without spinal maps of MN activation. Additionally, the absence of a reference database for locomotion makes the interpretation and comparison of results more challenging in clinical use. Even though some algorithms, such as NNMF, have been used for synergy extractions by a large number of groups, the interpretation and comparison of results also depend on small differences in application of the same basic methodology, and variability across groups in the results obtained when performing the same analysis has been observed. The software will allow the users, also with no expertise in the assessment of spinal muscle coordination patterns, to test their exoskeleton according to a consistent methodology and to have an index on how the exoskeleton perturbs the operator’s control strategy and affects/improves the spinal muscle coordination output in patients. Nevertheless, some experience in EMG data recording is required. Thus, while the PEPATO software provides an automatic detection of some types of artefacts (that allowed us to clean a part [even though small] of the EMG data, see Results), a caution should be taken prior to the experiments to ensure the best placement of the EMG electrodes and data recording to minimize potential artefacts, especially when using powered exoskeletons.

Various exoskeletons (passive, fully powered, with different degrees of freedom, assisted as needed, etc.) and control policies are being developed [1,3,4,5,6,7,8] and their control implementations can benefit from appropriate procedures and metrics for their evaluation [9,10]. For some exoskeletons, low sensitivity of some performance indicators, when compared with normal walking, can be related to non-significant changes due to ‘natural’ adaptation and/or a better matching between the subject’s locomotor output and the control strategy of a particular exoskeleton. However, the previous studies on neurologically intact individuals walking in the exoskeleton revealed some differences [21,23,28,29] as well as the differences in the spinal locomotor output are highly expected for patients [17,34,38,62].

In the provided example of using the PEPATO software for assessing walking in the unloading exoskeleton (Figure 3), the results revealed several significant changes in the spinal locomotor output (Figure 4 and Figure 5). An interesting finding consisted in specific alterations in the basic temporal patterns (CoA, FWHM) and also their differential changes, so that some motor modules were affected more than others for this particular exoskeleton. For instance, the most affected module seems to be module 4 (Figure 4c), loaded mainly on hamstring muscles and related to the control of the swing phase (Figure 4b) since (1). The 4th motor module was the least common for participants walking in the exoskeleton and participants during normal walking (see Results), and (2), its temporal pattern varying significantly across the subjects (Figure 4c upper panel and 4d right panel, see also a variable ensemble-averaged pattern in Figure 4a, lower panel). This result seems to corroborate our previous findings on walking in the powered exoskeleton showing that the most variable EMG activity was observed in the hamstring muscles [21]. Four basic locomotor modules are associated with major biomechanical functions during gait [13,41] (Figure 4b) and their differential impairment may help to get insights into specific impairments/adaptations during human–exoskeleton interactions. In contrast to the computational approach of assessing the motor modules based on common regularities in muscle activation, the spinal maps approach is based on an evolutionary adopted MN grouping of different muscles in the spinal cord and the corresponding motor pool activation loci [50]. The analysis of the spinal maps highlighted differential changes in the activation of sacral and lumbar segments (Figure 5b), which might be important for understanding the neural substrates and principles of the differential involvement of sacral and lumbar motor pools in the control and adaptation of human gait [53,63]. Locomotor movements can be accommodated to various external conditions and human–exoskeleton interactions in particular, and some of the suggestions in this article based on the analysis of the spinal locomotor output may possibly be revised or generalized as empirical data on the human–exoskeleton accumulated locomotor behaviour.

## 5. Conclusions

### Finding’s Remarks and Future Work

Exoskeleton robotic devices are now often used in the rehabilitation practice to assist physical therapy. Here we presented the PEPATO software, which is a benchmarking solution to assess changes in the spinal locomotor output during walking in the exoskeleton and is integrated at EUROBENCH facilities (https://eurobench2020.eu/, accessed on 28 May 2022). An example of using this SW for evaluating walking in the unloading exoskeleton demonstrated specific alterations in the basic temporal patterns related to the ‘impaired’ control of the swing phase (Figure 4) and specific changes in the spinal motor output (Figure 5). The proposed performance outcomes for evaluating changes in the neural control strategy based on the evaluation of muscle coordination patterns during walking in an exoskeleton will allow to perform the tests in any laboratory settings and compare the results with various exoskeletons. The PEPATO SW also envisages to create the custom reference database/values (that can be generated automatically using only the number of pre-processed trials stored in a standard format) to allow future users to have multiple reference data sets and to select the option based on the specific application at hand. In addition to medical applications for gait assistance and rehabilitation in patients, exoskeletons are being developed also for other applications for healthy subjects (military, industry, agriculture, etc.). Since market opportunities may be related to broad application of the exoskeletons, people who develop exoskeletons and want to test them may benefit from incorporating our methodology as an integral part of testing protocols. This will allow to test lower limb exoskeletons developed for various purposes and will provide indicators determining the variation of the spinal locomotor output during walking in such exoskeletons with respect to reference values.

## Figures and Tables

**Figure 1 sensors-22-05708-f001:**
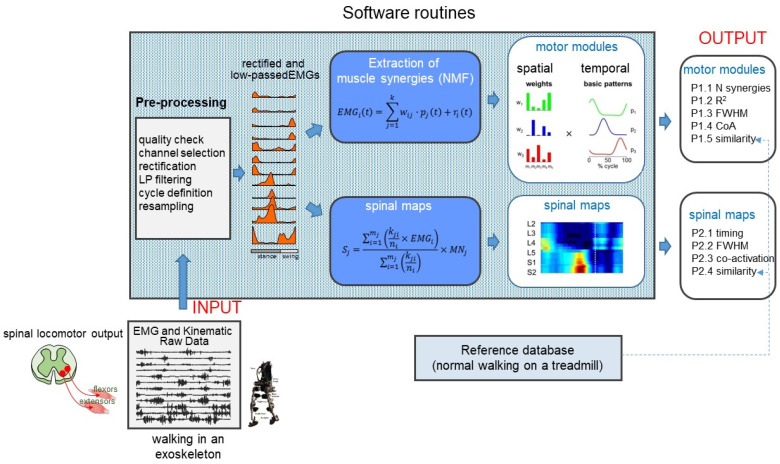
Overview of the spinal locomotor output evaluation system for walking in the exoskeleton. The acquired EMG signals and kinematic events serve as an input, and two main groups of performance indicators of the spinal muscle coordination output are generated as an output.

**Figure 2 sensors-22-05708-f002:**
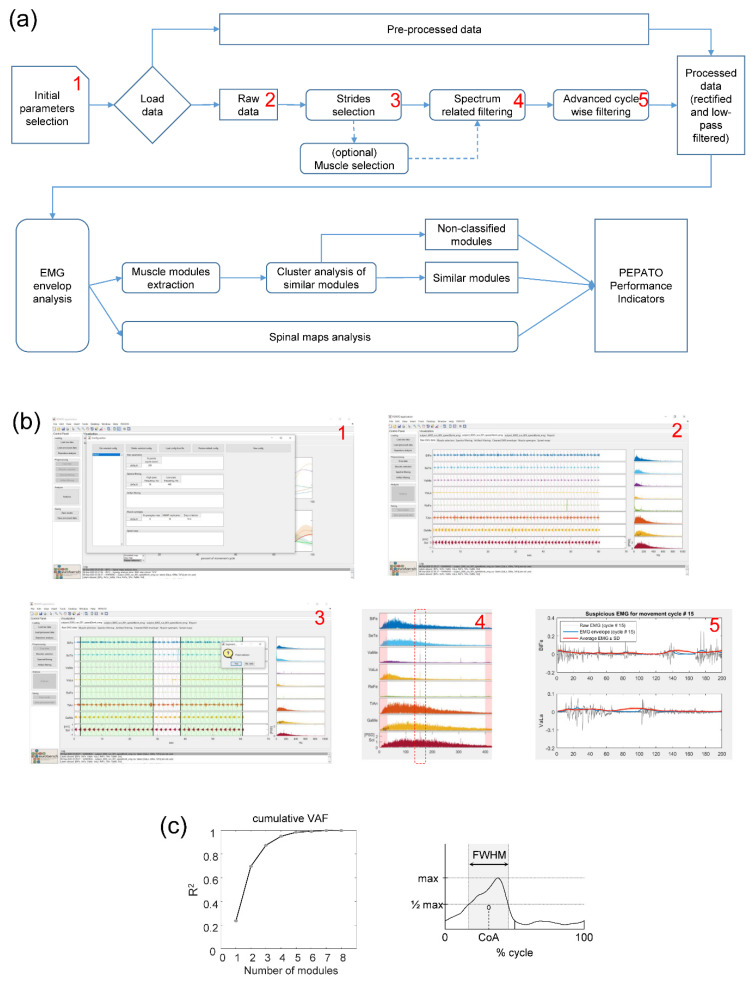
Pipeline and interfaces. (**a**) Pipeline of the PEPATO software for the EMG data analysis. (**b**) Examples of interfaces (screenshots) for the main elements of the data pre-processing chain: initial parameters selection (1), visualization of EMGs along with gait cycle timing and EMG spectra (2), exclusion of gait cycles with potential artefacts (3), identification of spectrum-related artefacts and their correction (4), warning for potential artefacts based on advanced artefact detection in individual cycles and their exclusion (5). (**c**) Cumulative VAF and FWHM, used for assessing the number of basic modules and the relative duration of basic pattern activity, accordingly. FWHM was calculated as the duration of the interval (in percent of gait cycle) in which EMG activity exceeded half of its maximum.

**Figure 3 sensors-22-05708-f003:**
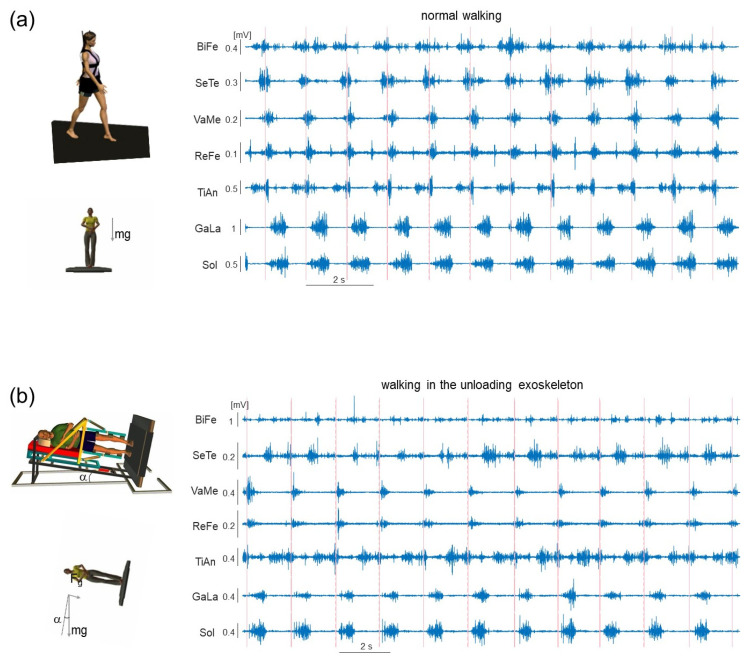
Examples of the EMG activity recordings during normal walking (**a**) and walking in the unloading exoskeleton (**b**) at 4 km/h. Experimental setup for recording of walking in the unloading exoskeleton is schematically shown on the left. Vertical lines correspond to the touchdown events in order to mark the individual gait cycles.

**Figure 4 sensors-22-05708-f004:**
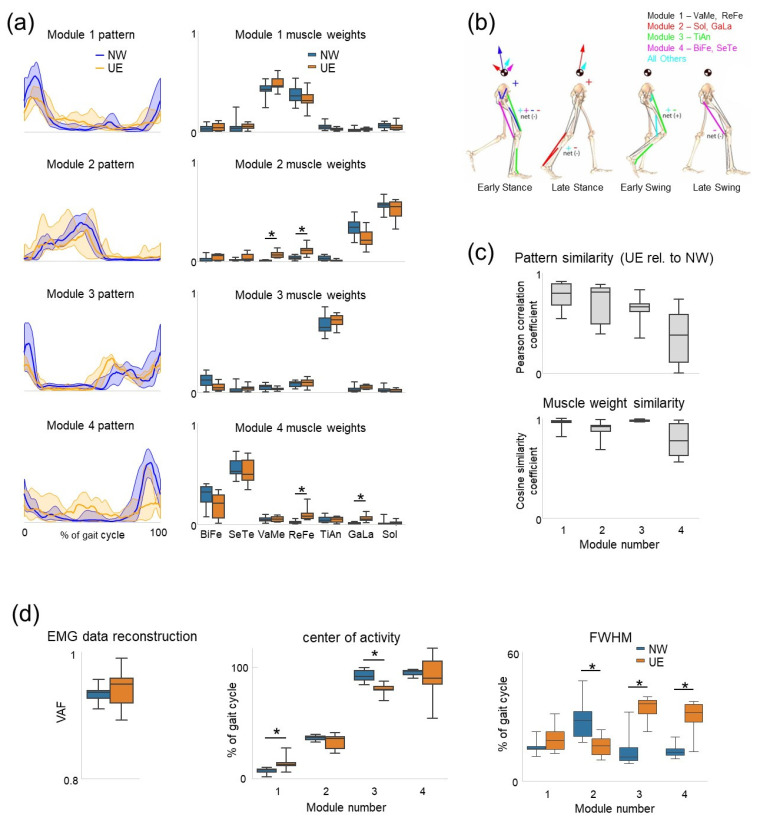
Assessment of performance indicators related to motor modules during walking in the unloading exoskeleton (UE) with respect to normal walking (NW). (**a**) Ensemble-averaged basic temporal patterns (±SD, left column) and corresponding muscle weights (right column, median and quartiles) of the two groups of subjects with four modules assumed for each group. Basic patterns were plotted in a “chronological” order (with respect to the timing of the main peak) vs. normalized gait cycle. (**b**) Biomechanical considerations: contributions of 4 basic patterns to the walking sub-tasks of body support, forward propulsion and leg swing during NW (modified from [40] with permission from Elsevier). Early stance (~15% of gait cycle), late stance (~45%), early swing (~70%), and late swing (~85%) are shown. Arrows departing from the center-of-mass denote the resultant module contributions to the horizontal and vertical ground reaction forces that accelerate the center-of-mass providing body support and forward propulsion. Net energy flow by each module to the trunk or leg is denoted by a + or - for energy increases or decreases, respectively. (**c**) Basic patterns similarity between UE and NW and similarity of corresponding muscle weights. (**d**) EMG data reconstruction (R^2^) using 4 modules, CoA of basic patterns, and FWHM. * Significant group differences.

**Figure 5 sensors-22-05708-f005:**
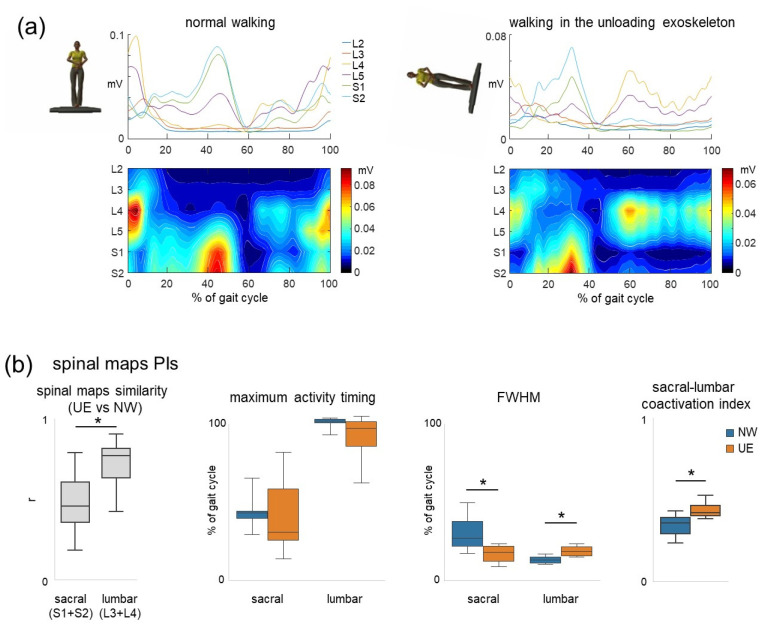
Assessment of spatiotemporal maps of MN activity in the lumbosacral enlargement during walking in the unloading exoskeleton (UE) with respect to normal walking (NW). (**a**)—Examples of individual maps during normal upright walking (on the left) and walking in the unloading exoskeleton (on the right). Output pattern of each segment is shown in the top panels, while the same pattern is plotted in a color scale at the bottom. The pattern is reported in mV (since the EMG signal from each muscle was expressed in mV, see Materials and Methods). Motor output (averaged across several strides) is plotted as a function of gait cycle and spinal segment level (L2–S2). (**b**)—Performance indicators for the spinal maps (from left to right): similarities of sacral (S1 + S2) and upper lumbar (L3 + L4) segment activity during walking in UE with respect to NW, timing of maximum activity in the sacral and upper lumber segments, FWHM, and coactivation index. The values are plotted as median and quartiles. * Significant group differences are marked by asterisks.

## Data Availability

Data on walking in the unloading exoskeleton are available upon reasonable request.

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
