# Peer review of "Evaluation of Spatiotemporal Patterns of the Spinal Muscle Coordination Output during Walking in the Exoskeleton"

_sensors, 2022, doi:10.3390/s22155708_

Round 1
Reviewer 1 Report
1. This article sounds focus on the PEPATO software, but framework of the full text is quite messy and redundant, which make the article lack of focus . May be because authors tried to put too many information in one article. Strongly suggest reconstruction and simplifying the whole paper, choose two points from the function introduction, math model and experiment analysis, and give a clear and consecutive narrative
2. The description of the content needs to be progressive one to another. Take Figure 1 as sample, the description of Figure 1 is in the section two, but two quotes in section one which make the reading pretty hard.
3. No chapter overview at the end of introduction
4. Most of the figures are too small
5. Lack of the comparison with other similar software or methodology.
Author Response
- This article sounds focus on the PEPATO software, but framework of the full text is quite messy and redundant, which make the article lack of focus. May be because authors tried to put too many information in one article. Strongly suggest reconstruction and simplifying the whole paper, choose two points from the function introduction, math model and experiment analysis, and give a clear and consecutive narrative
We thank the reviewer for his/her suggestions served to improve the manuscript. As suggested, we tried to give a clear narrative. In particular, we shortened and simplified some parts of the paper (e.g., we shortened the abstract by about 4 lines and the introduction by about 15 lines). We believe that our study on the assessments of changes in the spinal locomotor output during walking in the exoskeleton fits well with the scope of this special issue of Sensors on ‘Quantifying, Understanding and Improving Human-Exoskeleton Interaction’ related to the exoskeleton benchmarking metrics and protocols. We also modified the Introduction, providing a clear overview of the material (see our response below). Changes in the ms are marked in red.
- The description of the content needs to be progressive one to another. Take Figure 1 as sample, the description of Figure 1 is in the section two, but two quotes in section one which make the reading pretty hard.
As suggested, we moved Figure 1 to the section two, and modified the text accordingly, as well as we checked the quotes and description of other figures in the text.
- No chapter overview at the end of introduction
As suggested, we added an overview of the material at the end of the Introduction:
“Here we present the PEPATO software (SW), which is a benchmarking solution to assess changes in the spinal locomotor output during walking in the exoskeleton. The proposed outcomes may provide important information about changes in the neural control strategy and spinal locomotor output, that will complement other performance indicators and enrich evaluation capabilities of wearable exoskeletons and their users. First, we describe the protocol, SW pipeline and initial data pre-processing (section 2.1.1 and 2.1.2). The detailed description of the performance indicators is provided in sections 2.1.3 (motor modules) and 2.1.4 (spinal maps). Finally, we present an example of usage of this SW (section 2.2 and 3) for evaluating walking in an unloading exoskeleton [26] that allows a person to step in simulated reduced (Moon’s) gravity. The reason for using the unloading exoskeleton was to demonstrate the wide variety of tasks in which the SW can be applied. Simulated reduced gravity represents a well-controlled technique to study body-weight unloading and is widely used in gait rehabilitation as a tool to facilitate locomotor activity in individuals with neuromotor disorders, such as spinal cord injury, stroke, Parkinson disease, multiple sclerosis, cerebral palsy, etc. [27].”
- Most of the figures are too small
We increased the size of characters in the figures and slightly increased the size of figures (please consider that, according to the policy of the journal, we must use the template of the paper with the pre-defined size of the text and figures). Likely, the production editor will adjust the final size of the figures if necessary.
- Lack of the comparison with other similar software or methodology.
In the revised manuscript we reported (page 16):
“Current approaches and available methodology to evaluate muscle coordination patterns during locomotion are mainly based on selected and limited parameters of multi-muscle activity and, to our knowledge, there is no SW that comprehensively assesses both muscle modules and spinal maps of MN activation. Even though some algorithms, such as NNMF, have been used for synergy extractions by a large number of groups, the interpretation and comparison of results also depend on small differences in application of the same basic methodology, and variability across groups in the results obtained when performing the same analysis has been observed. The software will allow the users, also with no expertise in the assessment of spinal muscle coordination patterns, to test their exoskeleton according to a consistent methodology and to have an index on how the exoskeleton perturbs the operator’s control strategy and affects/improves the spinal muscle coordination output in patients.”
Reviewer 2 Report
1. The author lacks certain details in data input and collection. For example, In Page 4, what device is the data collected from? What is the frequency of the data? What is the specific process of data collection at different speeds? For example, is there an interval in each test? How often do you collect complete data? All of these are related to the integrity of the article.
2. In Page 4, the author does not elaborate on many details, for example:
Why 2km/h,4km/h, and 6km/h are selected? Is there any basis for this?
Why did the author choose these 8 muscles as measurement objects?
Why choose age gradients 19.9-36.4, 65.0-80.1? and why is there no record between 36.5-65?
3. FIG. 2 (a) is meaningless in my opinion. The picture is fuzzy and I can't see the content clearly. I suggest the author delete this figure or replace it with a clearer flow chart.
4. The author puts forward a software in the whole paper, but there is no corresponding detail description about it in the paper. I suggest that the author present the clear software interface and describe it in combination with its functions.
5. Is the software system designed for real-time analysis, and if so, what is the time delay for each function?
6. Author presents an example for evaluating walking in an unloading exoskeleton that allows a person to step in simulated reduced (Moon’s) gravity by tilting the body relative to the vertical. Why make example in this situation? Is there any consideration in that?
Author Response
We thank the reviewer for his/her comments and suggestions served to improve the manuscript. We tried to incorporate them all. Changes in the ms are marked in red.
- The author lacks certain details in data input and collection. For example, In Page 4, what device is the data collected from? What is the frequency of the data? What is the specific process of data collection at different speeds? For example, is there an interval in each test? How often do you collect complete data? All of these are related to the integrity of the article.
As suggested, we added the details for the reference data collection in page 4 of the revised manuscript:
“Each walking trial lasted ~40 s and was performed on a standard treadmill. The intervals between trials were ~1 min. Prior to data collection, the subjects were trained on each task, allowing them time to familiarize with the various walking conditions. EMG data were recorded at 2000 Hz using a Delsys Trigno wireless system (Boston, MA). Kinematic data were recorded at 100 Hz by means of a Vicon-612 system (Oxford, UK) with nine cameras. Sampling of kinematics and EMG data was synchronized. For each trial, at least 10 full gait cycles were recorded, and no more than 10% of them were dropped due to the artefacts of different nature (evoked by movement, electrode contact interruption).”
- In Page 4, the author does not elaborate on many details, for example:
Why 2km/h,4km/h, and 6km/h are selected? Is there any basis for this?
Why did the author choose these 8 muscles as measurement objects?
Why choose age gradients 19.9-36.4, 65.0-80.1? and why is there no record between 36.5-65?
As suggested, we added some details:
Line 131: “This set of speeds evenly covers the common range of normal human walking speeds, from a slow walk (2 km/h) to almost a run (6 km/h), and thus can serve as a reference for exoskeleton walking.”
Line 144: “This set of muscles represent major extensor and flexors groups of muscles controlling walking and often used in both simulation studies and when evaluating basic muscle modules [40–42]. Also, these muscles constitute a relatively large part of the total cross-sectional area of lower limb muscles [43].”
Line 173: “Aging adults (65-80 yrs) may also serve as a control group for the populations of patients with an older age (e.g., Parkinson disease, stroke, etc.). Participants ranging from 40 to 65 yrs were not included in order to sharpen the age differences between the two groups, nevertheless, the PEPATO SW allows data reference updating.”
In addition, we had previously indicated that:
(line 154): “For initial benchmarking of the PEPATO SW, we created two sets of reference data according to the specifications provided below. However, future users of the SW may want to create their own set of reference data based on the specific characteristics of the population and conditions to be tested, e.g. age range, sex, typicality or pathology, normal walking or walking with an exoskeleton, speed range, etc.”
and line 188: “PEPATO allows data reference updating to include more muscles, subjects and conditions.”
- FIG. 2 (a) is meaningless in my opinion. The picture is fuzzy and I can't see the content clearly. I suggest the author delete this figure or replace it with a clearer flow chart.
As suggested, we replaced Figure 2 with a flow chart.
- The author puts forward a software in the whole paper, but there is no corresponding detail description about it in the paper. I suggest that the author present the clear software interface and describe it in combination with its functions.
As suggested, we included a pipeline with interfaces in the new figure 2. We also provided an overview of the reported SW and the study at the end of the Introduction. The detailed description of the performance indicators is provided in sections 2.1.3 (motor modules) and 2.1.4 (spinal maps).
- Is the software system designed for real-time analysis, and if so, what is the time delay for each function?
Currently, the SW is designed for the off-line analysis. We added (ln 101): “Only offline processing is available, the SW is not configured for real-time analysis.”
- Author presents an example for evaluating walking in an unloading exoskeleton that allows a person to step in simulated reduced (Moon’s) gravity by tilting the body relative to the vertical. Why make example in this situation? Is there any consideration in that?
We added (line 89): “The reason for using an unloading exoskeleton was to demonstrate the wide variety of tasks in which the SW can be applied. Simulated reduced gravity represents a well-controlled technique to study body-weight unloading and is widely used in gait rehabilitation as a tool to facilitate locomotor activity in individuals with neuromotor disorders, such as spinal cord injury, stroke, Parkinson disease, multiple sclerosis, cerebral palsy, etc. [40].”
Reviewer 3 Report
I understand the findings of the present article, however, there are many insufficient explanations to be rethink as shown by the attachment.

Author Response
I understand the findings of the present article, however, there are many insufficient explanations to be rethink as follows.
We thank the reviewer for his/her evaluation of the study and suggestions served to improve the manuscript. Changes in the ms are marked in red.
- All figures are illegible enough to be incomprehensible because of diminutive delineation. I recommend the captions are magnified and the characters are changed boldface.
As suggested, we increased the size of characters in the figures and slightly increased the size of figures. (As for the figure captions, please take into account that, according to the policy of the journal, we cannot change the font for the figure legends since we must use the template of the paper with the pre-defined size of the text. Also, the production editor will likely adjust the final size of figures if necessary.
- Module 1, 2, 3, 4 are insufficient explanation. For examples, can other additional diagrams be prepared?
As suggested, we added a new subplot (b) in Figure 4 with biomechanical considerations related to the contribution of basic patterns (modules) to the walking biomechanics. Thank you for this suggestion.
- The abscissa of Fig. 4 is incomprehensible. Why is it from 0 to 1, for example, in (b)? What is “% cicle”, for example, in (e)?
We increased the size of characters in the figure. We added the measures of similarity from 0 to 1 in (c) (old panel (b)): ‘Pearson correlation coefficient’ and ‘Cosine similarity coefficient’. We also added to the figure legend that EMG data are plotted “vs. normalized gait cycle” and we replaced “% cycle” with “% of gait cycle” in all figures.
- The ordinate and the abscissa of Fig. 5(a) are incomprehensible.
We increased the size of characters in the figure. We also added to the figure legend that ”the pattern is reported in mV (since the EMG signal from each muscle was expressed in mV, see Materials and Methods)”.
Others:
- “Abstract” is lengthy. Be revised.
As suggested, we shortened the abstracts (by about 3.5 lines).
- “Conclusion” should include the finding’s remarks by the present study to be summarized as well as the comments of future work. Be revised.
As suggested, we also added the finding’s remarks to Conclusions: “An example of using this SW for evaluating walking in the unloading exoskeleton demonstrated specific alterations in the basic temporal patterns related to the ‘impaired’ control of the swing phase (Figure 4) and specific changes in the spinal motor output (Figure 5)”.
Round 2
Reviewer 1 Report
The comparison with other similar software or methodology is without sufficient valid information,for example, technical differences and corresponding effects。
Author Response
The comparison with other similar software or methodology is without sufficient valid information,for example, technical differences and corresponding effects。
As suggested, for comparison, we added an example of publicly available SW (page 16, line 582-587, marked in red):
"For example, the SW musclesyneRgies [61] offers a complete analysis framework of muscle synergies addressed to scientists and more advanced users, with customizable sensible defaults depending on the specifics of the study design, but without spinal maps of MN activation. Also, the absence of reference database for locomotion makes the interpretation and comparison of results more challenging in clinical use."
and related reference:
"61. Santuz, A. musclesyneRgies: factorization of electromyographic data in R with sensible defaults. Journal of Open Source Software, 2022, 7(74),
4439. https://doi.org/10.21105/joss.04439."
Reviewer 3 Report
It's OK.
Author Response
Thank you again for your comments and evaluation of the ms.